# Arctic Sea Ice Microalga *Chlamydomonas latifrons* KNF0041: Identification and Statistical Optimization of Medium for Enhanced Biomass and Omega-3/Omega-6

**DOI:** 10.3390/md21080454

**Published:** 2023-08-17

**Authors:** Hyunsik Chae, Sanghee Kim, Han-Gu Choi, Ji Hee Kim, Se Jong Han, Eun Jae Kim

**Affiliations:** 1Division of Life Sciences, Korea Polar Research Institute, Incheon 21990, Republic of Korea; chaehs85@kopri.re.kr (H.C.);; 2Department of Polar Sciences, University of Science and Technology, Incheon 21990, Republic of Korea

**Keywords:** *Chlamydomonas latifrons*, identification, medium optimization, biomass enhancement

## Abstract

Polar microorganisms produce biologically active compounds that enable them to survive in harsh environments. These compounds have potential biomedical applications. The green microalga *Chlamydomonas latifrons* KNF0041, isolated from Arctic sea ice, has been found to produce polyunsaturated fatty acids (PUFAs), including omega-3 and omega-6, which have antioxidant properties. To improve the biomass production of strain KNF0041, statistical methods such as the Plackett–Burman design, Box–Behnken design, and response surface methodology (RSM) were utilized for medium optimization. The optimized medium was designed with increased potassium phosphate content and reduced acetic acid (AcOH) content. The use of the optimized medium resulted in an increase in the cell number as biomass of strain KNF0041 by 34.18% and the omega-3 and omega-6 fatty acid (FA) content by 10.04% and 58.29%, respectively, compared to that in normal TAP medium, which is known as the growth medium for *Chlamydomonas* culture. In this study, *Chlamydomonas latifrons* was discovered for the first time in the polar region and identified using morphology and molecular phylogenetic analyses, the secondary structures of the internal transcribed spacers, and optimized culture conditions. The results of this study provide an efficient method for the application of polar microalgae for the production of bioactive compounds.

## 1. Introduction

The polar regions, including Antarctica and the Arctic, are characterized by persistently cold climates. These extreme environments have given rise to unique adaptations in the organisms inhabiting these regions. In particular, microalgae have developed various strategies to survive harsh conditions, such as the production of cold-active enzymes and ice-binding proteins [1,2,3,4]. These adaptations allow them to cope with low temperatures and protect themselves from the detrimental effects of ice crystal formation [5]. In addition, microalgae in polar regions accumulate polyunsaturated fatty acids (PUFAs) to maintain membrane fluidity and counteract the physical damage caused by freezing [6].

In this study, we focused on exploring the biodiversity of microalgae in the polar regions. Through expeditions to the Arctic, we discovered the microalga *Chlamydomonas latifrons* Nygaard (strain KNF0041). Genetic analysis revealed a close relationship with the well-studied model species *C. reinhardtii* Dangeard. The genus *Chlamydomonas* Ehrenberg is a large group of unicellular green algae consisting of over 400 described species based on morphological characteristics [7]. However, phylogenetic studies have shown that the traditional concept of this genus is polyphyletic [8,9]. To address this issue, taxonomic revisions and phylogenetic analyses are underway to establish a more accurate classification system for *Chlamydomonas* species.

The strain KNF0041, the psychrotrophic microalga that we discovered, exhibits the ability to thrive at cold temperatures, making it suitable for cultivation in cold regions or during winter. This strain overcomes the typical low productivity associated with mesophilic strains at low temperatures, making it promising for various applications, such as in the production of biofuels, biomaterials, cosmetics, and biomedicine. Furthermore, its close genetic resemblance to *C. reinhardtii* offers opportunities for the genetic manipulation and engineering of desirable traits in microalgae. Notably, the strain KNF0041 produces a high proportion of omega-3 and omega-6 highly unsaturated fatty acids, which are essential in maintaining overall health [10].

Polar microalgae, including the *Chlamydomonas* species, have attracted interest as potential sources of PUFAs. PUFAs such as linoleic acid, alpha/gamma-linolenic acid (ALA/GLA), eicosapentaenoic acid (EPA), and docosahexaenoic acid (DHA) have various applications in the food industry as additives and nutritional supplements [6,11]. Additionally, extracts from microalgae, such as *Chloromonas reticulata* (Gorozhankin) Gobi KSF0100 and *Micractinium* sp. KSF0105, isolated from Antarctica, have shown anticancer and anti-inflammatory properties, while *Micractinium variabile* H. Chae, H.-G. Choi, & J.H. Kim KSF0031 produces significant amounts of phytol, which exhibits antiviral effects [12,13].

The primary objective of this study was to clarify the phylogenetic relationships within the phylogroup *Reinhardtinia* through the characterization of the strain KNF0041. Additionally, we aimed to optimize the growth conditions and enhance the production of omega-3/omega-6 fatty acids in the Arctic microalga *C. latifrons* KNF0041. Microalgal growth can be easily enhanced by selecting an appropriate culture medium, pH, temperature, light intensity, and other factors. However, combining these traditional methods in experiments to find conditions that maximize growth can be challenging. For example, traditional methods of changing one variable at a time while keeping others constant can be time-consuming and overlook potential interactions among factors. Therefore, we employed statistical experimental designs, such as a two-factorial design and the response surface methodology, to identify the optimal medium components to promote the growth of *C. latifrons* KNF0041. By applying these statistical tools, we aimed to improve our understanding of the effective medium composition and establish efficient conditions for the production of omega-3/omega-6 fatty acids in this microorganism. The results of this study can be utilized to effectively produce secondary metabolites and bioactive compounds from microalgae found in polar regions. 

## 2. Results and Discussion 

### 2.1. Morphological Characteristics of the Strain KNF0041

Cells of strain KNF0041 isolated from the Arctic region (Figure 1), herein identified as *Chlamydomonas latifrons*, were spherical, broadly ellipsoidal, ellipsoidal, or often ovoid (Figure 2), 11.0–19.1 μm long and 9.3–16.6 μm wide (Appendix A). The cell wall was thin and contained a prominent anterior papilla (Figure 2A–C,F). Two flagella of equal length, approximately 1–1.5 times as long as the cell (Figure 2A), and two contractile vacuoles were present below the flagellar bases (Figure 2A–C,F). The cells possessed single cup-shaped chloroplasts with surface slits (Figure 2E). The single pyrenoid was large, spherical, or broadly ellipsoidal, surrounded by several starch grains, and located in a thick part of the chloroplast (Figure 2A–C). The nucleus was located in the anterior part of the cell (Figure 2C). An ellipsoidal eyespot was observed at the anterior position (Figure 2D). Asexual reproduction via zoospore formation involved two or four zoospores (Figure 2F). The morphology of the zoospores was similar to that of vegetative cells. Sexual reproduction was not observed. *Chlamydomonas latifrons* KNF0041 exhibited morphological characteristics similar to the other *Chlamydomonas* species, but the presence of a papilla was observed only in *C. latifrons* (Appendix A). To visualize the intracellular lipids of the cells cultured for 4 weeks, Nile Red staining was performed, and fluorescence microscopy revealed the presence of numerous lipid droplets emitting yellow-orange fluorescence (Appendix A). 

### 2.2. Molecular Phylogenetic Analyses

Phylogenetic analysis of the nuclear small-subunit (SSU) rDNA revealed that *Chlamydomonas latifrons* KNF0041 was placed within the phylogroup *Reinhardtinia* (Figure 3). *C. latifrons* KNF0041 clustered as a part of a supported subclade containing *C. reinhardtii*, the *Chlamydomonas* type species. The subclade consisted of three strains (KNF0024 and KNF0041 as *C. latifrons*, SAG 70.81 as *C.* cf. *latifrons*). The two strains of *C. latifrons*, KNF0024 and KNF0041, differed by one nucleotide from *C.* cf. *latifrons* SAG 70.81.

For a deeper analysis, we used combined SSU and internal transcribed spacer (ITS) rDNA datasets. Three species of *Chlamydomonas* (*C. reinhardtii*, *C. incerta*, and *C. schloesseri*) were sisters to *C. latifrons* KNF0041 and *C.* cf. *latifrons* SAG 70.81 (Figure 4). *C. latifrons* KNF0041 was identical in its SSU and ITS sequences (2427 aligned nucleotides) and differed from *C.* cf. *latifrons* SAG 70.81 by 47 nucleotide substitutions. 

### 2.3. Secondary Structures of ITS1 and ITS2

To confirm that strain KNF0041 was *C. latifrons*, we analyzed the secondary structures of ITS1 and ITS2 to detect the compensatory base changes (CBCs) and hemi-CBCs (Figure 5 and Figure 6). These structures and base changes differed from those of closely related *Chlamydomonas* strains. The secondary structures of ITS1 of the five *Chlamydomonas* strains were similar, except for helix III of *C. latifrons* KNF0041 and *C.* cf. *latifrons* SAG 70.81 (Figure 5*). C. latifrons* KNF0041 differed from *C.* cf. *latifrons* SAG 70.81 in two CBCs and three hemi-CBCs. The base pair differences in the conserved ITS1 region of the five *Chlamydomonas* strains were two–nine CBCs and two–three hemi-CBCs (Figure 5). Figure 6 shows the predicted secondary structure of the ITS2 of *C. latifrons* KNF0041, which was compared to that of other strains of *Chlamydomonas*. *C. latifrons* KNF0041 differed from *C. reinhardtii* SAG 11-32b, the type species of the genus, in terms of three CBCs and three hemi-CBCs. *C. latifrons* KNF0041 was closely related to *C.* cf. *latifrons* SAG 70.81 but differed by one hemi-CBC in helix II of ITS2 (Figure 6).

### 2.4. Determination of Growth Temperature, Medium, and Light Conditions

To determine the optimal culturing temperature for strain KNF0041, we conducted cultivation experiments at various temperatures, including 4, 8, 12, and 20 °C. After a 15-day culture period in Bold’s Basal Medium (BBM), we observed distinct growth patterns at different temperatures. The highest cell yield, reaching 3.0 × 10^6^ cells/mL, was achieved at 8 °C (Figure 7A), while its growth was relatively slower at 4 °C. In other words, the strain KNF0041 thrives in the temperature range of 8 °C to 20 °C. This indicates that strain KNF0041 falls under the category of psychrophilic or psychrotrophic microorganisms, exhibiting robust growth at low temperatures below 20 °C, which suggests its adaptation to cold environments [14]. The growth characteristics of the strain KNF0041 were evaluated in three selective media: F/2 (representing seawater), Tris-Acetate-Phosphate (TAP), and BBM (representing freshwater). Although the strain KNF0041 demonstrated growth in both seawater and freshwater media, the highest cell density was observed when cultured in TAP medium (Figure 7B). The effect of light on KNF0041 was examined by exposing the samples to various light-emitting diode (LED) light intensities, including 40, 80, 120, and 160 μmol photon m^−2^ s^−1^. The maximum cell concentration was observed at an intensity of 80 μmol photon m^−2^ s^−1^ (Figure 7C). Exposure to light intensities exceeding 160 μmol photon m^−2^ s^−1^ appeared to be detrimental to the growth of KNF0041, potentially disrupting the photosynthesis system [15].

### 2.5. Component Selection 

To eliminate unnecessary or negative elements in the microalgal medium, all seven components were analyzed individually (Figure 8). The composition of the TAP medium used in this experiment was as follows: Tris base, 2.42 g/L; NH_4_Cl, 0.375 g/L; MgSO_4_·7H_2_O, 0.1 g/L; CaCl_2_·2H_2_O, 0.05 g/L; potassium phosphate, 0.432 g/L (K_2_HPO_4_, 0.288 g/L and KH_2_PO_4_, 0.144 g/L); 1 mL of acetic acid; 1 mL of trace metals containing Na_2_EDTA·2H_2_O, 50 g/L; ZnSO_4_·7H_2_O, 22 g/L; H_3_BO_3_, 11.4 g/L; MnCl_2_·4H_2_O, 5 g/L; FeSO_4_·7H_2_O, 5 g/L; CoCl_2_·6H_2_O, 1.6 g/L; CuSO_4_·5H_2_O, 1.6 g/L; and (NH_4_)6MoO_3_, 1.1 g/L. Using TAP medium as the base medium, an elimination method analysis was performed by individually removing each component of the medium to identify the essential ingredients. The results revealed that all components had positive effects, with *p*-values below 0.05, indicating their significance in promoting growth (Figure 8). 

### 2.6. Screening of Components Using the Plackett–Burman Design

The components that had a positive effect were identified using the Plackett–Burman method. For this purpose, the concentrations of the seven key components were either increased to twice (+value) their original levels or reduced to one fifth (−value) of their original levels to precisely determine the effective ingredients (Table 1). Among the components examined, levels of trace metals were non-significant (*p* > 0.05). Owing to their significance in the element elimination experiment, trace metals were used in the final medium at the original concentration instead of being removed. Essentially, trace metals are essential components for cell growth [16]. The Tris base, NH_4_Cl, MgSO_4_, and AcOH had negative effects, whereas CaCl_2_ and potassium phosphate had positive effects (Table 1).

### 2.7. Optimization of Medium Components for KNF0041 Growth

A Box–Behnken design was employed to explore the optimal conditions considering the interaction effects. In this analysis, six effective components identified using the Plackett–Burman method were included, taking into account their positive and negative effects. The two components that exhibited positive effects (CaCl_2_ and potassium phosphate) were set at twice their original concentrations, whereas the four components that showed negative effects (Tris base, NH_4_Cl, MgSO_4_, and AcOH) were maintained at their original concentrations (Table 2). The Box–Behnken design method was used to investigate the interactions among these components. A total of 54 experiments were conducted (including three replicates of each of 18 experiments), in which the concentrations of the six components were adjusted to three levels, −value, 0 value, and +value, and the interactions between two components were evaluated. The interactions were represented as follows: Tris base (*X*_1_) and NH_4_Cl (*X*_2_), Tris base (*X*_1_) and MgSO_4_ (*X*_3_), Tris base (*X*_1_) and CaCl_2_ (*X*_4_), Tris base (*X*_1_) and potassium phosphate (*X*_5_), Tris base (*X*_1_) and AcOH (*X*_6_), NH_4_Cl (*X*_2_) and MgSO_4_ (*X*_3_), NH_4_Cl (*X*_2_) and AcOH (*X*_6_), NH_4_Cl (*X*_2_) and potassium phosphate (*X*_5_), NH_4_Cl (*X*_2_) and CaCl_2_ (*X*_4_), MgSO_4_ (*X*_3_) and CaCl_2_ (*X*_4_), MgSO_4_ (*X*_3_) and potassium phosphate (*X*_5_), MgSO_4_ (*X*_3_) and AcOH (*X*_6_), CaCl_2_ (*X*_4_) and potassium phosphate (*X*_5_), CaCl_2_ (*X*_4_) and AcOH (*X*_6_), potassium phosphate (*X*_5_) and AcOH (*X*_6_). The effects of the six variables at different concentrations are shown in Figure 9 and Appendix A, which show the three-dimensional response and contour plots. The predicted cell densities were calculated as follows: [23 − 42*X*_1_ + 461*X*_2_ + 404*X*_3_ + 7404*X*_4_ + 85*X*_5_ − 80*X*_6_ − 32*X*_1_^2^ − 1838*X*_2_^2^ − 5655*X*_3_^2^ − 62157*X*_4_^2^ − 97*X*_5_^2^ − 468*X*_6_^2^ + 382*X*_1_*X*_2_ − 19*X*_1_*X*_3_ + 64*X*_1_*X*_4_ + 3*X*_1_*X*_5_ + 251*X*_1_*X*_6_ + 3742*X*_2_*X*_3_ + 256*X*_2_*X*_4_ − 7*X*_2_*X*_5_ + 28*X*_2_*X*_6_ − 2167*X*_3_*X*_4_ − 38*X*_3_*X*_5_ − 42*X*_3_*X*_6_ − 1320*X*_4_*X*_5_ − 51*X*_4_*X*_6_ + 265*X*_5_*X*_6_] × 10^4^ cells/mL, (*X*_1_ = Tris base, *X*_2_ = NH4Cl, *X*_3_ = MgSO_4_, *X*_4_ = CaCl_2_, *X*_5_ = potassium phosphate, and *X*_6_ = AcOH). 

The optimal composition and concentration of the culture medium for the microalgal strain KNF0041, derived from the aforementioned analysis, are shown in Table 3. Under these conditions, the cell density of the microalgae was calculated to be 5.8 × 10^6^ cells/mL (Figure 10). The optimized composition of the medium was as follows: Tris base, 2.42 g/L; NH_4_Cl, 0.375 g/L; MgSO_4_·7H_2_O, 0.1 g/L; CaCl_2_·2H_2_O, 0.05 g/L; potassium phosphate, 0.864 g/L (K_2_HPO_4_, 0.288 g/L and KH_2_PO_4_, 0.144 g/L); 0.81 mL of AcOH; 1 mL of trace metals containing Na_2_EDTA·2H_2_O, 50 g/L; ZnSO_4_·7H_2_O, 22 g/L; H_3_BO_3_, 11.4 g/L; MnCl_2_·4H_2_O, 5 g/L; FeSO_4_·7H_2_O, 5 g/L; CoCl_2_·6H_2_O, 1.6 g/L; CuSO_4_·5H_2_O, 1.6 g/L; and (NH_4_)6MoO_3_, 1.1 g/L (Table 3). The amount of potassium phosphate was doubled, and the amount of AcOH decreased from 1 mL to 0.81 mL, whereas the remaining components remained unchanged. The phosphorus present in potassium phosphate plays a significant role in the growth of microalgae, influencing carbon and nitrogen utilization, as well as contributing to lipid regulation [17]. Furthermore, potassium phosphate acts as a buffer to maintain the pH levels of the medium [18].

### 2.8. Comparison of the Strain KNF0041’s Growth and Omega-3/Omega-6 Fatty Acid Production in Normal and Optimized Media

Cultivating KNF0041 in the optimized medium composition revealed a significant difference in cell density compared to normal TAP medium. The largest difference in cell density (34.18%) was observed on the 21st day. The optimized TAP medium yielded a cell density of 5.55 × 10^6^ cells/mL on the 28th day, which closely matched the predicted values (5.8 × 10^6^ cells/mL) obtained through statistical analysis (Figure 11A). 

The strain KNF0041 is classified as a psychrotrophic or psychrophilic microalga, as it thrives well at 8 °C [1,6,19]. Psychrophilic microalgae are known for their high cellular accumulation of unsaturated fatty acids in their cells [20,21]. Table 4 presents the fatty acid composition of the Arctic microalga KNF0041, with C18:3 (20.45%), C16:4 (12.07%), and C18:4 (10.1%) being omega-3 and omega-6 polyunsaturated fatty acids. 

The most abundantly produced fatty acids in *C. latifrons* KNF0041 were omega-3 ALA, reaching 41.55 mg/g DCW. According to a study reported by Zheng et al. [22], the temperature and light conditions were optimized to enhance the production of ALA in *C. reinhardtii*, reaching approximately 20.53 mg/g DCW (equivalent to 18.87 mg/L/d when converted to mg/g). According to a study reported by Kim et al. [19], when optimizing the light and aeration conditions and inducing nitrogen deficiency, linolenic acid production in Arctic *Chlamydomonas* sp. KNF0008 reached 47 mg/g DCW. Although the present study did not incorporate aeration and nitrogen deficiency treatments, there is the potential to further enhance the productivity of secondary metabolites by implementing such conditions in future studies. When KNF0041 was cultivated in the optimized medium, a significant increase in omega-3 and omega-6 fatty acid content was observed (Figure 11B). Using the optimized medium, the production of omega-3 fatty acids reached 97.69 mg/g DCW, whereas that of omega-6 fatty acids reached 38.20 mg/g DCW. This represented a 10.04% increase in omega-3 fatty acid production and a 58.29% increase in omega-6 fatty acid production compared to normal TAP medium (Figure 11B). The total fatty acid production in the optimized TAP medium was 183.13 mg/g DCW, showing a 15.65% increase compared to that in the normal TAP medium. Methods to increase the intracellular omega-3 fatty acid content, such as adjusting the salinity or inducing a nitrogen deficiency, have been reported. According to a study by Tsai et al. [23], employing both methods resulted in an approximately 3% increase in omega-3 fatty acid content in *Tetraselmis* sp. DS3. According to a report by Caliskan and Haznedaroglu [24], nitrogen deficiency during the cultivation of *Chlorococcum novae-angliae* P.A. Archibald & Bold resulted in a 31.84% increase in omega-3 fatty acid production. However, such stress conditions were found to enhance the intracellular fatty acid content, but did not benefit cell growth [23,24]. In this study, the statistical design of the medium for *C. latifrons* KNF0041 increased the biomass production, as predicted. The Plackett–Burman design and Box–Behnken design utilized in this study as methods to optimize the medium can be further applied to research aimed at maximizing the production of specific secondary metabolites, such as phytol, similar to the case of *M. variabile* KSF0031 [13]. These optimization methods have significant potential for the industrialization of microalgae that produce valuable secondary metabolites, thereby providing essential approaches for their commercialization. 

## 3. Materials and Methods

### 3.1. Isolation and Culture Condition

The strain KNF0041 was isolated from sea ice near the Arctic Dasan station in Ny-Ålesund, Spitsbergen, Norway (78° 56.667′ N, 11° 45.571′ E) (Figure 1). Individual cells of the strain were isolated using a sterile Pasteur capillary pipette and transferred to a 12-well plate. Unialgal isolates were cultured in Bold Modified Basal Freshwater nutrient solution liquid medium (B5282; Sigma-Aldrich, St. Louis, MO, USA) at approximately 2 °C. The culture was maintained in BBM at 2–3 °C with a light intensity of 20–30 μmol photons m^−2^ s^−1^ provided by cool-white fluorescent lamps, following a 16:8 h light–dark cycle. The strain KNF0041 was deposited in the Korea Polar Research Institute (KOPRI) Culture Collection for Polar Microorganisms.

Growth experiments were conducted under static conditions at 4, 8, 12, and 20 °C, using cool white fluorescent lamps (80 µmol photon m^−2^ s^−1^) on a 16:8 h light–dark cycle. To determine a suitable medium and temperature for growth, the samples were cultured in F/2, BBM, and TAP medium [6]. The optimal light intensity was assessed by inoculating 2.5 × 10^5^ cells/mL in BBM and measuring cell growth under static conditions at 8 °C with various white fluorescent lights (40, 80, 120, and 160 μmol photon m^−2^ s^−1^).

### 3.2. Morphological and Molecular Identification and Phylogenetic Analyses

Cells were counted using a hemocytometer and morphological investigations of the strain were performed using an Axio Imager A2 (Carl Zeiss, Oberkochen, Germany) equipped with Nomarski differential interference contrast optics. Microscopic images were captured using the AxioCam HRC camera (Carl Zeiss, Oberkochen, Germany). The morphological characteristics were described as outlined in Ettl et al. [7] and Pröschold et al. [25].

For lipid staining, Nile Red dye (9-(diethylamino)benzo[a]phenoxazin-5(5H)-one, Sigma-Aldrich, Australia) was dissolved in acetone, and the final cell treatment concentration was 1 μg/mL. Lipid bodies were analyzed using fluorescence microscopy (Axio Imager A2; Carl Zeiss, Oberkochen, Germany). The Nile Red dye signal, indicating the presence of lipids, was captured by laser excitation at 488 nm, whereas chlorophyll fluorescence was detected by laser excitation at 633 nm.

Genomic DNA was extracted from the algal cells using an i-Genomic Plant DNA Extraction Kit (iNtRON Biotechnology, Seoul, Republic of Korea) according to the manufacturer’s protocol. The nuclear SSU, ITS1, 5.8S, and ITS2 rDNA were amplified by polymerase chain reactions (PCRs) using the primer pairs NS1/NS4, NS5/NS8, and ITS1/ITS4 [26]. The PCR products were purified using the MG PCR Product Purification Kit (Macrogen, Seoul, Republic of Korea) according to the manufacturer’s protocol and sent to Macrogen (Seoul, Republic of Korea) for sequencing.

Phylogenetic analyses were performed separately for two datasets: (i) a concatenated dataset of 39 SSU rDNA sequences, and (ii) 16 SSU and ITS rDNA sequences. Sequences were aligned using BioEdit 7.0.5.3 [27]. The GenBank accession numbers for all the sequences used are shown in Figure 3 and Figure 4. For all datasets, an appropriate evolutionary model was calculated using jModelTest 2 [28] under the Akaike information criterion, and the GTR + I + G nucleotide substitution model was selected as the best model for maximum likelihood (ML) and Bayesian inference (BI). A phylogenetic tree was constructed by ML analysis using PhyML v3.0 [29] with 1000 bootstrap replicates. BI was performed using MrBayes 3.2.6 [30] with 1,000,000 generations of Markov chain Monte Carlo iterations, and the first 25% of the trees were discarded as burn-ins. Phylogenetic trees were visualized using FigTree v1.4.2 (available at http://tree.bio.ed.ac.uk/software/figtree/, accessed on 30 June 2023).

### 3.3. ITS1 and ITS2 Secondary Structures

For species delimitation, the secondary structures of the ITS1 and ITS2 regions were reconstructed. Secondary structure models of ITS1 and ITS2 from strain KNF0041 (OR250719) were folded using Mfold [31] based on the minimum energy criterion and visualized using PseudoViewer3 (http://pseudo-viewer.sha.ac.kr/, accessed on 22 June 2023). These models were manually refined, and the reference models were developed by Pröschold et al. [25].

### 3.4. Statistical Optimization

To determine the essential components for the growth of the strain KNF0041, a Plackett–Burman design [32] was utilized. The Plackett–Burman design is a well-established approach to the efficient screening of large variables, requiring only n + 1 experiments to identify the critical parameters [33]. Based on these results, five elements in the TAP medium were selected and represented at two levels, low (−) and high (+) concentrations, resulting in a first-order model. The effect of each factor on the cell number was calculated using Equation (1):*Y* = β0 + Σβi*X*i(1)
where *Y* represents the response (number of cells), β0 is the model intercept, βi is the linear factor coefficient, and *X*i is the level of each variable. The concentrations of the medium components were optimized using the Box–Behnken surface response method [34]. The Box–Behnken design is suitable for the analysis of factors that vary among the three levels (−1, 0, +1) [35]. In this study, individual factors were coded into three distinct levels: low (−), intermediate (0), and high (+). A second-order polynomial model, represented by Equation (2), was used to assess the influence of the individual components on the predicted response:*Y* = β0 + Σβi*X*i + Σβij*X*i*X*j + Σβii*X*i^2^(2)

In Equation (2), Y denotes the predicted response (number of cells), β0 is the constant coefficient, βi, βij, and βii are the regression coefficients of the model, and *X*i and *X*j represent the coded values of independent nutrient variables.

### 3.5. Fatty Acid Methyl Ester (FAME) Analysis

For fatty acid extraction, 20 mg of the freeze-dried KNF0041 sample was used. Fatty acid methyl esters (FAMEs) were obtained from crude lipids and analyzed as previously described [6,36]. FAMEs were quantified using an external standard (1 mg of C22:0 in hexane). FAMEs were analyzed by gas chromatography (YL-6100GC, Young Lin Science, Republic of Korea) with a flame-ionized detector (FID) and an Omegawax 250 capillary column (30 m × 0.25 mm × 0.25 μm, Supelco, Bellefonte, PA, USA). Gas chromatography–mass spectrometry (GC-MS) was performed using a TSQ 8000 EVO GC-MS instrument (Thermo Fisher Scientific, Waltham, MA, USA), and the results were confirmed using authentic standards (Supelco 37 Component FAME Mix and phytol from Sigma-Aldrich). The FAME content (*w*/*w*, %) and yield (mg/L) were determined and expressed as milligrams of FAMEs per gram of dry cell weight (DCW) and milligrams of FAMEs per liter of culture volume, respectively.

### 3.6. Statistical Tests

Statistical significance was determined based on a *p*-value of <0.05. The optimization values for the significant medium components were calculated using the Minitab software, version 14 (Minitab Inc., State College, PA, USA), and linear multiple regression was employed for statistical analysis.

## 4. Conclusions

The strain KNF0041, the first discovery of *Chlamydomonas latifrons* in the Arctic region, was identified using the latest microalgal identification techniques based on the morphology, molecular phylogeny, and the secondary structures of ITS1 and ITS2 rDNA sequences. Using RSM, we optimized the cell growth and enhanced the productivity of valuable substances such as PUFAs and omega-3/omega-6 fatty acids in this psychrophilic microalga. Further research focusing on optimizing the production of valuable compounds would be beneficial. Through these studies, we anticipate additional discoveries of psychrophilic microalgae in polar regions, the increased production of biologically valuable substances using these microalgae, and expanded industrial applications of microalgal products.

## Figures and Tables

**Figure 1 marinedrugs-21-00454-f001:**
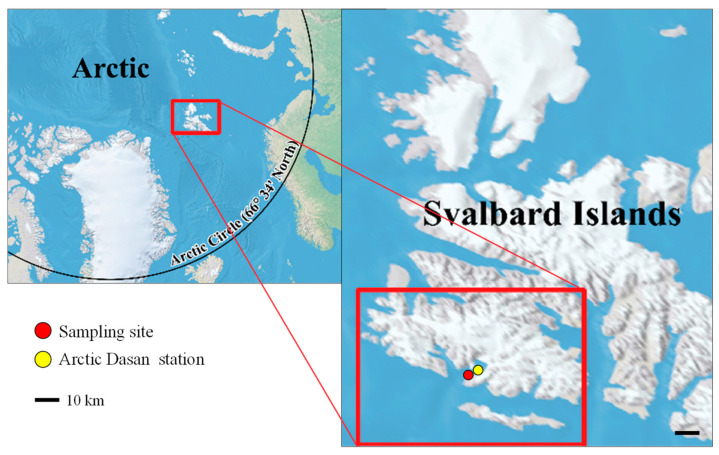
Collection location. The strain KNF0041 was isolated from sea ice near the Arctic Dasan station in Ny-Ålesund, Svalbard Islands, in the Arctic.

**Figure 2 marinedrugs-21-00454-f002:**
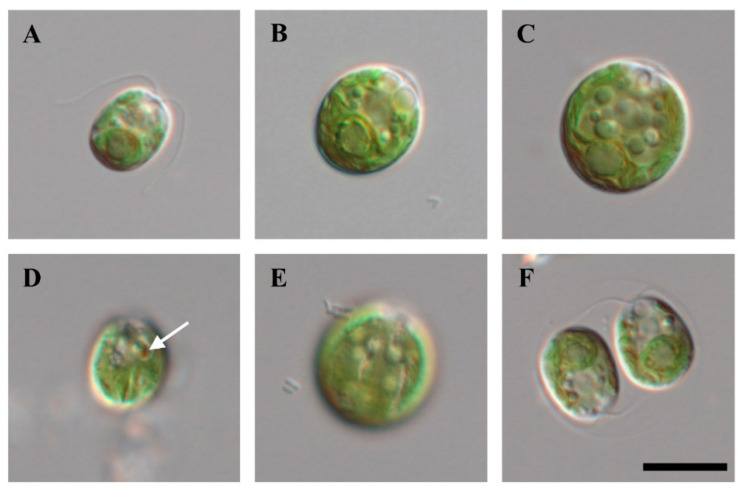
Light microscopy of strain KNF0041. (**A**–**C**) Vegetative cells, (**D**) marked eyespot, (**E**) surface view of cells with slits in the chloroplast, (**F**) zoosporangium with two daughter cells. Scale bar = 10 μm.

**Figure 3 marinedrugs-21-00454-f003:**
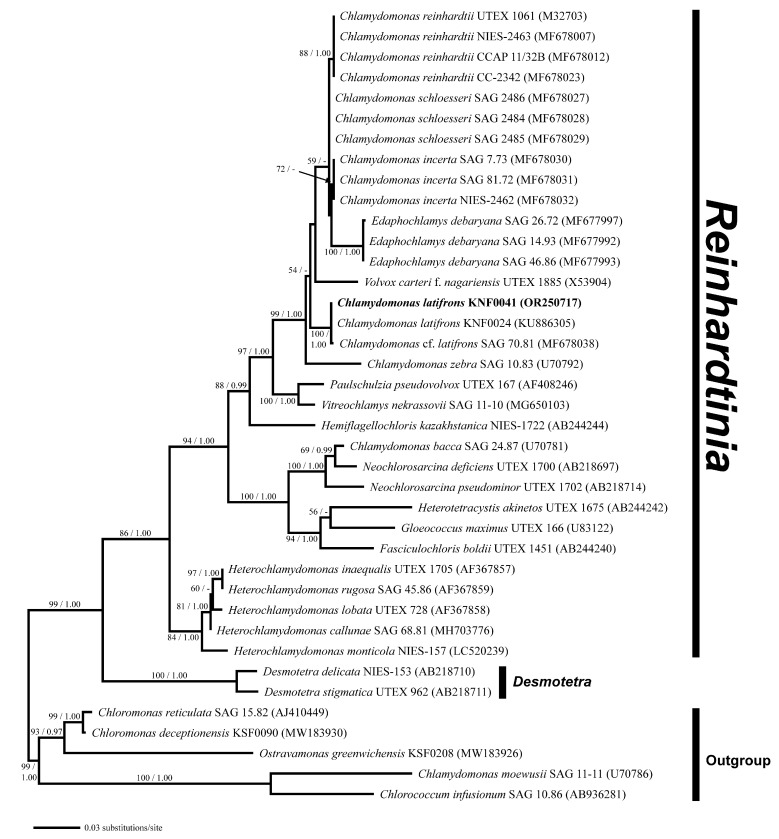
Maximum likelihood (ML) tree constructed from small-subunit ribosomal DNA sequences. The numbers at each node are the ML (>50%, left) and Bayesian probability (>0.95, right). The new sequence is shown in bold.

**Figure 4 marinedrugs-21-00454-f004:**
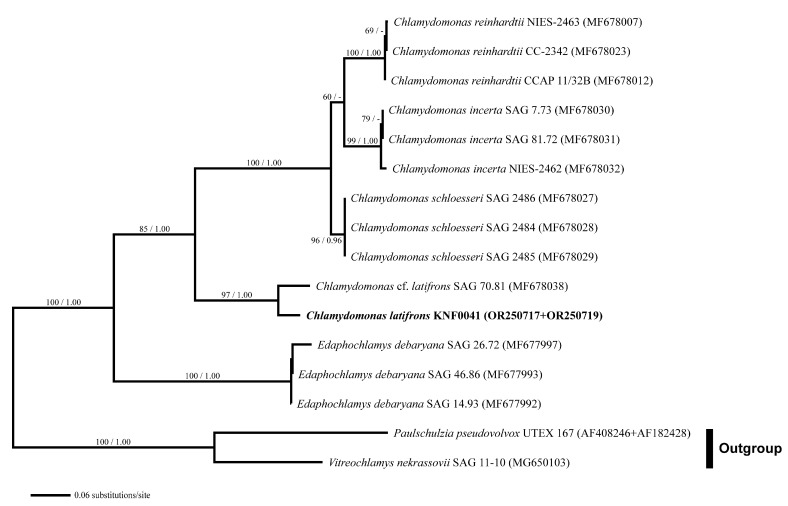
Maximum likelihood (ML) tree constructed from the small-subunit, internal transcribed spacer 1, 5.8S, and internal transcribed spacer 2 ribosomal DNA sequences. The numbers at each node are the ML (>50%, left) and Bayesian probability (>0.95, right). The new sequences are shown in bold.

**Figure 5 marinedrugs-21-00454-f005:**
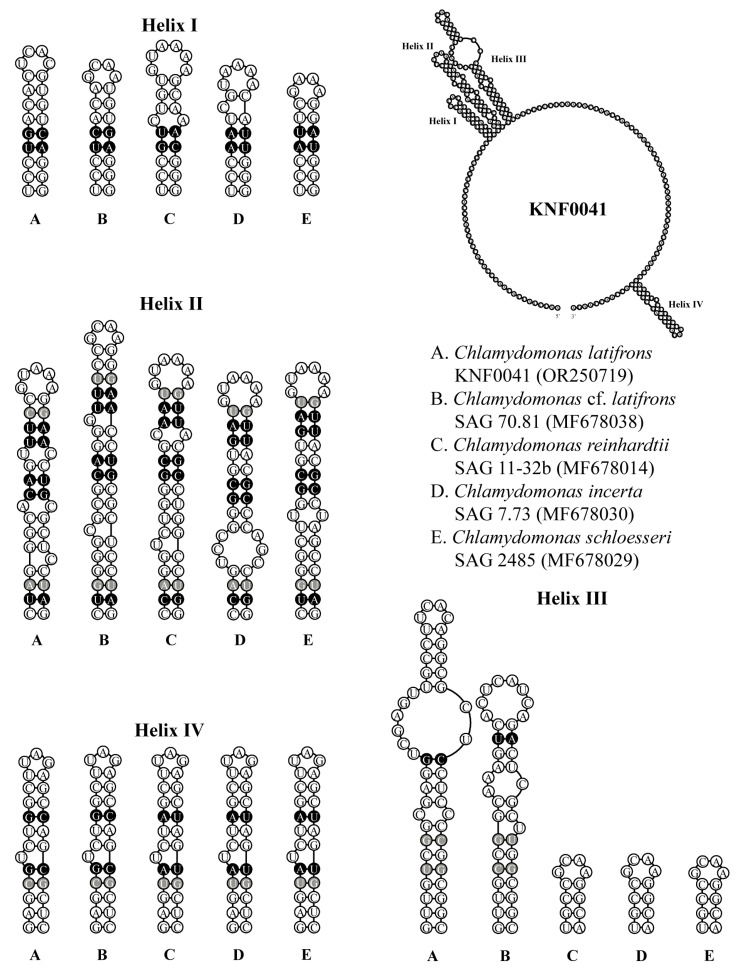
Secondary structure diagrams of internal transcribed spacer 1 ribosomal DNA sequences among *Chlamydomonas* species. The compensatory base changes (CBCs) and hemi-CBCs are marked in black and grey, respectively.

**Figure 6 marinedrugs-21-00454-f006:**
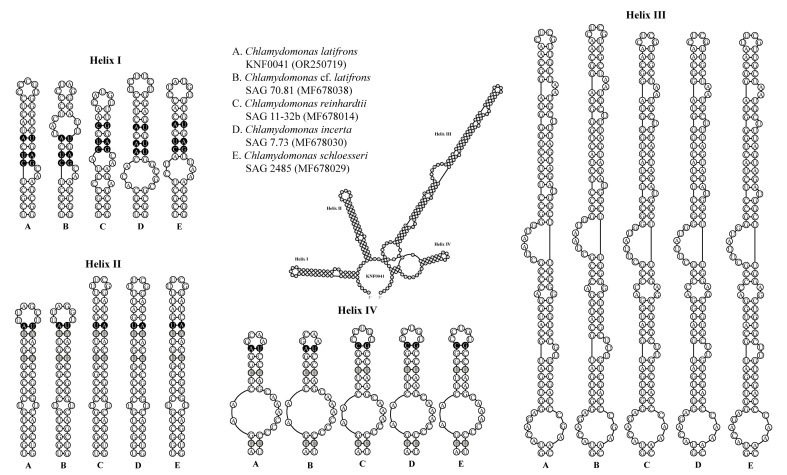
Secondary structure diagrams of internal transcribed spacer 2 ribosomal DNA sequences among *Chlamydomonas* species. The compensatory base changes (CBCs) and hemi-CBCs are marked in black and grey, respectively.

**Figure 7 marinedrugs-21-00454-f007:**
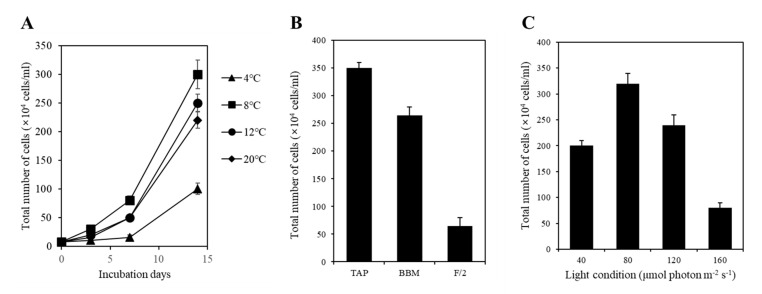
Tests of temperature, medium, and light intensity under culture conditions. (**A**) Growth curves of the strain KNF0041 at various temperatures. (**B**) The total number of cells cultured over a period of 2 weeks in different growth media, including F/2 as a seawater medium and BBM and TAP as freshwater media, was measured. (**C**) The total number of cells cultured under four different light conditions for 2 weeks was measured.

**Figure 8 marinedrugs-21-00454-f008:**
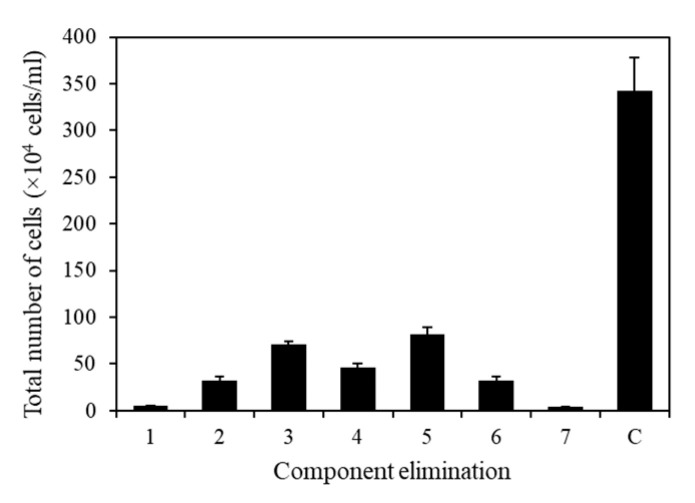
Effect of medium on KNF0041 cell growth. Total number of cells in the absence of 1, Tris base; 2, NH_4_Cl; 3, MgSO_4_·7H_2_O; 4, CaCl_2_·2H_2_O; 5, potassium phosphate; 6, trace metals; 7, acetic acid; and C, control (no absence). Data shown in this study represent the average cell densities ± respective standard deviation from triplicate measurements (*n* = 3).

**Figure 9 marinedrugs-21-00454-f009:**
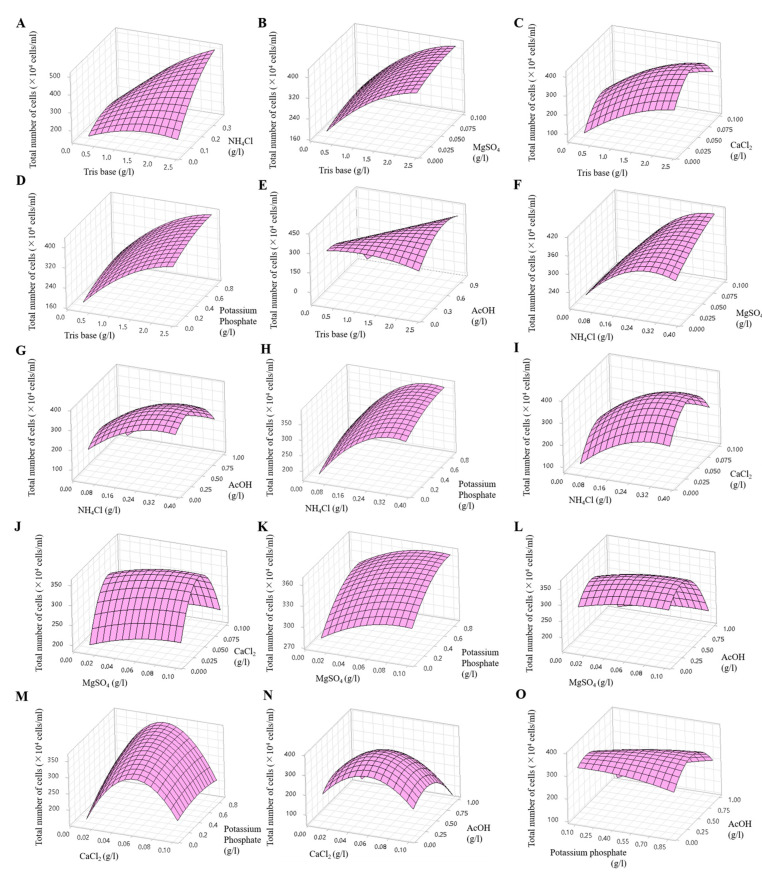
Three-dimensional response surface plots showing the effects of Tris base and NH_4_Cl (**A**), Tris base and MgSO_4_ (**B**), Tris base and CaCl_2_ (**C**), Tris base and potassium phosphate (**D**), Tris base and AcOH (**E**), NH_4_Cl and MgSO_4_ (**F**), NH_4_Cl and AcOH (**G**), NH_4_Cl and potassium phosphate (**H**), NH_4_Cl and CaCl_2_ (**I**), MgSO_4_ and CaCl_2_ (**J**), MgSO_4_ and potassium phosphate (**K**), MgSO_4_ and AcOH (**L**), CaCl_2_ and potassium phosphate (**M**), CaCl_2_ and AcOH (**N**), potassium phosphate and AcOH (**O**).

**Figure 10 marinedrugs-21-00454-f010:**
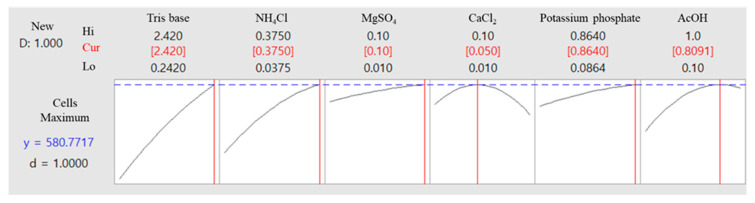
Optimal concentrations of the selected components were used to achieve maximum cell density and predict total cell numbers under these optimized conditions.

**Figure 11 marinedrugs-21-00454-f011:**
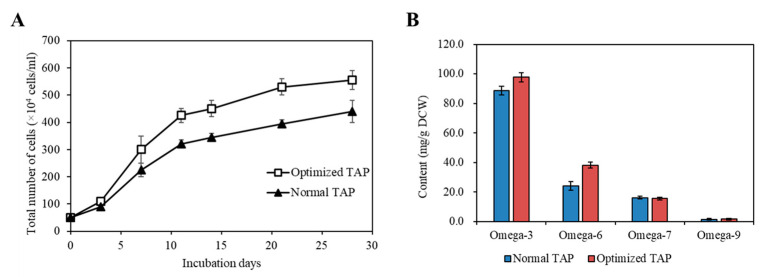
Comparison of the strain KNF0041’s biomass production under various conditions. (**A**) The growth of the strain KNF0041’s cells. The solid triangle represents the normal and the open squares represent the optimized TAP medium conditions. (**B**) Omega-3, -6, -7, -9 fatty acid yields of the strain KNF0041.

**Table 1 marinedrugs-21-00454-t001:** Statistical analyses of the medium components were performed using the Plackett–Burman test.

Variable	Component	−Value (g/L)	+Value (g/L)	Effect	*T*-Statistic	*p*-Value
*X* _1_	Tris base	0.484	4.84	−30.58	−5.54	0
*X* _2_	NH_4_Cl	0.075	0.75	−12.48	−2.26	0.036
*X* _3_	MgSO_4_	0.02	0.2	−65.72	−11.91	0
*X* _4_	CaCl_2_	0.01	0.1	30.93	5.6	0
*X* _5_	Potassium phosphate	0.0864	0.864	25.37	4.6	0
*X* _6_	Trace metals	0.01954	0.1954	−7.83	−1.42	0.173
*X* _7_	AcOH	0.2 (mL)	2 (mL)	−67.82	−12.29	0

**Table 2 marinedrugs-21-00454-t002:** Box–Behnken optimization of selected significant medium components.

Variable	Component	−Value (g/L)	0 Value (g/L)	Value (g/L)
*X* _1_	Tris base	0.242	1.331	2.42
*X* _2_	NH_4_Cl	0.0375	0.20625	0.375
*X* _3_	MgSO_4_	0.01	0.055	0.1
*X* _4_	CaCl_2_	0.01	0.055	0.1
*X* _5_	Potassium phosphate	0.0864	0.4752	0.864
*X* _6_	AcOH	0.1 (mL)	0.55 (mL)	1 (mL)

**Table 3 marinedrugs-21-00454-t003:** Medium composition of normal and optimized TAP.

Component	Normal TAP (g/L)	Optimized TAP (g/L)
Tris base	2.42	2.42
NH_4_Cl	0.375	0.375
MgSO_4_	0.1	0.1
CaCl_2_	0.05	0.05
Potassium phosphate	0.432	0.864
Trace metals	0.097	0.097
AcOH	1 (mL)	0.81 (mL)

**Table 4 marinedrugs-21-00454-t004:** Comparison of fatty acid yields, including omega-3/omega-6, from KNF0041.

Fatty Acid	Normal TAP		Optimized TAP	
	Content (%)	Yield (mg/g DCW)	Content (%)	Yield (mg/g DCW)
C12:0	0.51	1.18	0.50	1.29
C14:0	0.52	1.18	0.51	1.29
C16:0	10.29	23.61	10.01	25.35
C16:1n-7	7.06	16.18	6.15	15.58
C16:2n-6	2.27	5.20	2.95	7.46
C16:3n-6	2.45	5.63	2.59	6.56
C16:4n-3	11.95	27.41	12.07	30.56
C18:0	0.78	1.78	0.83	2.11
C18:1n-9	0.65	1.49	0.64	1.62
C18:2n-6	2.09	4.79	5.51	13.95
C18:3n-6	3.71	8.52	4.04	10.22
C18:3n-3	16.31	37.42	16.41	41.55
C18:4n-3	10.44	23.95	10.10	25.58
phytol	1.12	2.56	1.67	4.23
SFA	12.19	27.76	11.85	30.04
MUFA	7.70	17.67	6.79	17.20
PUFA	49.23	112.91	53.66	135.89
TFA	69.12	158.35	72.31	183.13

## Data Availability

Available on request.

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
