# Peer review of "Arctic Sea Ice Microalga Chlamydomonas latifrons KNF0041: Identification and Statistical Optimization of Medium for Enhanced Biomass and Omega-3/Omega-6"

_marinedrugs, 2023, doi:10.3390/md21080454_

Round 1
Reviewer 1 Report
In this manuscript, the authors presented a biomass production optimization method using a Plackett-Burman design, Box-Behnken design, and response surface methodology. The use of the optimized medium resulted in an increase in the cell number as biomass of Chlamydomonas latifrons KNF0041 and omega-3 and omega-6 fatty acid contents. The study has been well conducted and the manuscript has been well written in a good agreement with the scientific requirements of the journal. However, there are still some concerns about this study:
- Line numbers should be added to the manuscript for ease of review.
- Every abbreviation in the first time in the text should be written in full. Please note to this comment in the whole of manuscript. Avoid, where is possible, to use abbreviations.
- In section: 2.7. Optimization of medium components for KNF0041 growth: The results and discussion of the optimization are not clearly presented and should be rewritten, as a number of details are missing, and there is no discussion or references.
- Can this optimization method apply to other process and how?
- Page 8: 2.5. Component selection: Do you mean nine components or seven?
- In Figure 8, the title of the x-axis variable is missing.
- In Figure 9, the units of the x-axis variable are missing.
- Check all references in the text, since they are presented in different formats: Prِschold et al. (2018), Shiyan Zheng et al. (2022), Kim et al. (2019), Tsai et al. in 2016, Caliskan, Haznedaroglu (2022), and Ettl et al. (1976). Change these references to numbers as the rest of the manuscript, and rearrange the references in all manuscript. The authors should observe the guidance of this journal to adjust this manuscript.
Author Response
*Please see the attachment

Reviewer 2 Report
Page 2 – in the introduction the authors are certainly correct in stating that statistical methods for experimental design can be very effective in identifying the optimum conditions as well as the interactions between factors. However, in some cases they can have the downside that it is difficult to identify casual relationships between factors, something more readily achieved using traditional, one factor at a time methods. This issue should at least be acknowledged in their discussion for the sake of completeness.
Page 4 – for the sake of completeness the authors should define acronyms like CBC and ITS.
Figure 7 B and C – it would be useful to know at what time these values were obtained – i.e. how long were the cells cultivated for?
Page 9 – the authors should be aware that screening for trace metals can be complex, as the cells need these in very small quantities. This can easily be carried over from the medium used in the inoculation, and/or be present in the water used to prepare the medium (amongst other things). Unless the authors have taken careful steps to rule out these factors it may not be possible to conclude the absence of the trace metals solution is significant.
Author Response
*Please see the attachment
